# Space-time Mixing Attention for Video Transformer

**Adrian Bulat**
Samsung AI Cambridge
adrian@adrianbulat.com

**Juan-Manuel Perez-Rua**
Samsung AI Cambridge
j.perez-rua@samsung.com

**Swathikiran Sudhakaran**
Samsung AI Cambridge
swathikir.s@samsung.com

**Brais Martinez**
Samsung AI Cambridge
brais.a@samsung.com

**Georgios Tzimiropoulos**
Samsung AI Cambridge
Queen Mary University of London
g.tzimiropoulos@qmul.ac.uk

## Abstract

This paper is on video recognition using Transformers. Very recent attempts in this area have demonstrated promising results in terms of recognition accuracy, yet they have been also shown to induce, in many cases, significant computational overheads due to the additional modelling of the temporal information. In this work, we propose a Video Transformer model the complexity of which scales linearly with the number of frames in the video sequence and hence induces *no overhead* compared to an image-based Transformer model. To achieve this, our model makes two approximations to the full space-time attention used in Video Transformers: (a) It restricts time attention to a local temporal window and capitalizes on the Transformer's depth to obtain full temporal coverage of the video sequence. (b) It uses efficient space-time mixing to attend *jointly* spatial and temporal locations without inducing any additional cost on top of a spatial-only attention model. We also show how to integrate 2 very lightweight mechanisms for global temporal-only attention which provide additional accuracy improvements at minimal computational cost. We demonstrate that our model produces very high recognition accuracy on the most popular video recognition datasets while at the same time being significantly more efficient than other Video Transformer models. Code for our method is made available here.

## 1 Introduction

Video recognition – in analogy to image recognition – refers to the problem of recognizing events of interest in video sequences such as human activities. Following the tremendous success of Transformers in sequential data, specifically in Natural Language Processing (NLP) [39, 5], Vision Transformers were very recently shown to outperform CNNs for image recognition too [48, 13, 35], signaling a paradigm shift on how visual understanding models should be constructed. In light of this, in this paper, we propose a Video Transformer model as an appealing and promising solution for improving the accuracy of video recognition models.

A direct, natural extension of Vision Transformers to the spatio-temporal domain is to perform the self-attention *jointly* across all $S$ spatial locations and $T$ temporal locations. Full space-time attention though has complexity $O(T^2 S^2)$ making such a model computationally heavy and, hence, impractical even when compared with the 3D-based convolutional models. As such, our aim is to exploit the temporal information present in video streams while minimizing the computational burden within the Transformer framework for efficient video recognition.

35th Conference on Neural Information Processing Systems (NeurIPS 2021).

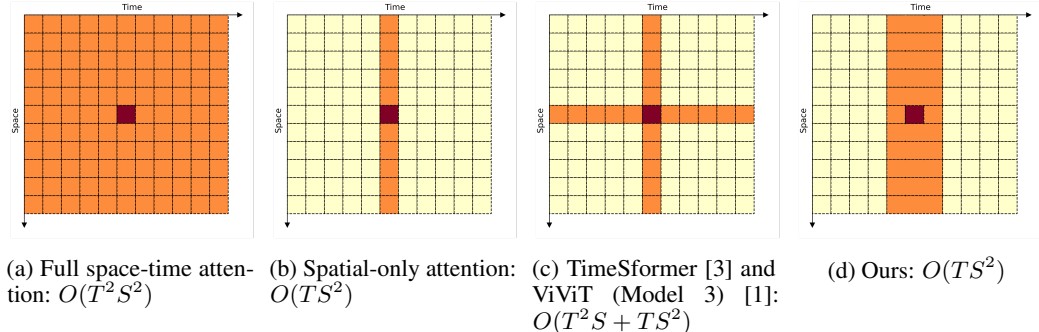

(a) Full space-time attention: $O(T^2 S^2)$    (b) Spatial-only attention: $O(TS^2)$    (c) TimeSformer [3] and ViViT (Model 3) [1]: $O(T^2 S + TS^2)$    (d) Ours: $O(TS^2)$

Figure 1: Different approaches to space-time self-attention for video recognition. In all cases, the key locations that the query vector, located at the center of the grid in red, attends are shown in orange. Unlike prior work, our key vector is constructed by mixing information from tokens located at the same spatial location within a local temporal window. Our method then performs self-attention with these tokens. Note that our mechanism allows for an efficient approximation of local space-time attention at no extra cost.

A baseline solution to this problem is to consider spatial-only attention followed by temporal averaging, which has complexity $O(TS^2)$. Similar attempts to reduce the cost of full space-time attention have been recently proposed in [3, 1]. These methods have demonstrated promising results in terms of video recognition accuracy, yet they have been also shown to induce, in most of the cases, significant computational overheads compared to the baseline (spatial-only) method due to the additional modelling of the temporal information.

**Our main contribution** in this paper is a Video Transformer model that has complexity $O(TS^2)$ and, hence, is as efficient as the baseline model, yet, as our results show, it outperforms recently/concurrently proposed work [3, 1] in terms of efficiency (*i.e.* accuracy/FLOP) by significant margins. To achieve this our model makes two approximations to the full space-time attention used in Video Transformers: (a) It restricts time attention to a local temporal window and capitalizes on the Transformer's depth to obtain full temporal coverage of the video sequence. (b) It uses efficient space-time mixing to attend *jointly* spatial and temporal locations without inducing any additional cost on top of a spatial-only attention model. Fig. 1 shows the proposed approximation to space-time attention. We also show how to integrate two very lightweight mechanisms for global temporal-only attention, which provide additional accuracy improvements at minimal computational cost. We demonstrate that our model is surprisingly effective in terms of capturing long-term dependencies and producing very high recognition accuracy on the most popular video recognition datasets, including Something-Something-v2 [17], Kinetics [4] and Epic Kitchens [9], while at the same time being significantly more efficient than other Video Transformer models.

## 2 Related work

**Video recognition:** Standard solutions are based on CNNs and can be broadly classified into two categories: 2D- and 3D-based approaches. 2D-based approaches process each frame independently to extract frame-based features which are then aggregated temporally with some sort of temporal modeling (e.g. temporal averaging) performed at the end of the network [42, 26, 27]. The works of [26, 27] use the "shift trick" [45] to have some temporal modeling at a layer level. 3D-based approaches [4, 16, 36] are considered the current state-of-the-art as they can typically learn stronger temporal models via 3D convolutions. However, they also incur higher computational and memory costs. To alleviate this, a large body of works attempt to improve their efficiency via spatial and/or temporal factorization [38, 37, 15].

**CNN vs ViT:** Historically, video recognition approaches tend to mimic the architectures used for image classification (e.g. from AlexNet [23] to [20] or from ResNet [18] and ResNeXt [47] to [16]). After revolutionizing NLP [39, 32], very recently, Transformer-based architectures showed promising results on large scale image classification too [13]. While self-attention and attention were previously used in conjunction with CNNs at a layer or block level [6, 50, 33], the Vision Transformer (ViT)

of Dosovitskiy et al. [13] is the first convolution-free, Transformer-based architecture that achieves state-of-the-art on ImageNet [11].

**Video Transformer:** Recently/concurrently with our work, vision transformer architectures, derived from [13], were used for video recognition [3, 1], too. Because performing full space-time attention is computationally prohibitive (*i.e.* $O(T^2 S^2)$), their main focus is on reducing this via temporal and spatial factorization. In TimeSformer [3], the authors propose applying spatial and temporal attention in an alternating manner reducing the complexity to $O(T^2 S + T S^2)$. In a similar fashion, ViViT [1] explores several avenues for space-time factorization. In addition, they also proposed to adapt the patch embedding process from [13] to 3D (*i.e.* video) data. Our work proposes a completely different approximation to full space-time attention that is also efficient. To this end, we firstly restrict full space-time attention to a local temporal window which is reminiscent of [2] but applied here to space-time attention and video recognition [1]. Secondly, we define a local joint space-time attention which we show that can be implemented efficiently via the "shift trick" [45].

## 3  Method

**Video Transformer:** We are given a video clip $\mathbf{X} \in \mathbb{R}^{T \times H \times W \times C}$ ($C = 3$). Following ViT [13], each frame is divided into $K \times K$ non-overlapping patches which are then mapped into visual tokens using a linear embedding layer $\mathbf{E} \in \mathbb{R}^{3K^2 \times d}$. Since self-attention is permutation invariant, in order to preserve the information regarding the location of each patch within space and time we also learn two positional embeddings, one for space: $\mathbf{p}_s \in \mathbb{R}^{1 \times S \times d}$ and one for time: $\mathbf{p}_t \in \mathbb{R}^{T \times 1 \times d}$. These are then added to the initial visual tokens. Finally, the token sequence is processed by $L$ Transformer layers.

The visual token at layer $l$, spatial location $s$ and temporal location $t$ is denoted as:

$$\mathbf{z}_{s,t}^l \in \mathbb{R}^d, \quad l = 0, \dots, L-1, \; s = 0, \dots, S-1, \; t = 0, \dots, T-1. \tag{1}$$

In addition to the $ST$ visual tokens extracted from the video, a special classification token $\mathbf{z}_{cls}^l \in \mathbb{R}^d$ is prepended to the token sequence [12]. The $l-$th Transformer layer processes the visual tokens $\mathbf{Z}^l \in \mathbb{R}^{(ST+1) \times d}$ of the previous layer using a series of Multi-head Self-Attention (MSA), Layer Normalization (LN), and MLP ($\mathbb{R}^d \to \mathbb{R}^{4d} \to \mathbb{R}^d$) layers as follows:

$$\mathbf{Y}^l = \text{MSA}(\text{LN}(\mathbf{Z}^{l-1})) + \mathbf{Z}^{l-1}, \tag{2}$$

$$\mathbf{Z}^l = \text{MLP}(\text{LN}(\mathbf{Y}^l)) + \mathbf{Y}^l. \tag{3}$$

The main computation of a single full space-time Self-Attention (SA) head boils down to calculating:

$$\mathbf{y}_{s,t}^l = \sum_{t'=0}^{T-1} \sum_{s'=0}^{S-1} \text{Softmax}\{(\mathbf{q}_{s,t}^l \cdot \mathbf{k}_{s',t'}^l)/\sqrt{d_h}\} \mathbf{v}_{s',t'}^l, \; \left\{ \begin{smallmatrix} s=0,\dots,S-1 \\ t=0,\dots,T-1 \end{smallmatrix} \right\} \tag{4}$$

where $\mathbf{q}_{s,t}^l, \mathbf{k}_{s,t}^l, \mathbf{v}_{s,t}^l \in \mathbb{R}^{d_h}$ are the query, key, and value vectors computed from $\mathbf{z}_{s,t}^l$ (after LN) using embedding matrices $\mathbf{W_q}, \mathbf{W_k}, \mathbf{W_v} \in \mathbb{R}^{d \times d_h}$. Finally, the output of the $h$ heads is concatenated and projected using embedding matrix $\mathbf{W_h} \in \mathbb{R}^{hd_h \times d}$.

The complexity of the full model is: $O(3hTSdd_h)$ ($qkv$ projections) $+O(2hT^2 S^2 d_h)$ (MSA for $h$ attention heads) $+O(TS(hd_h)d)$ (multi-head projection) $+O(4TSd^2)$ (MLP) [2]. From these terms, our goal is to reduce the cost $O(2T^2 S^2 d_h)$ (for a single attention head) of the full space-time attention which is the dominant term [3]. For clarity, from now on, we will drop constant terms and $d_h$ to report complexity unless necessary. Hence, the complexity of the full space-time attention is $O(T^2 S^2)$.

**Our baseline** is a model that performs a simple approximation to the full space-time attention by applying, at each Transformer layer, spatial-only attention:

$$\mathbf{y}_{s,t}^l = \sum_{s'=0}^{S-1} \text{Softmax}\{(\mathbf{q}_{s,t}^l \cdot \mathbf{k}_{s',t}^l)/\sqrt{d_h}\} \mathbf{v}_{s',t}^l, \; \left\{ \begin{smallmatrix} s=0,\dots,S-1 \\ t=0,\dots,T-1 \end{smallmatrix} \right\} \tag{5}$$

---

[1]Other attempts of exploiting local attention can be found in [29, 7, 49], however they are also different in scope, task/domain and implementation.

[2]For this work, we used $S = 196$, $T = \{8, 16, 32\}$ and $d = 768$ (for a ViT-B backbone).

[3]The MLP complexity is by no means negligible, however the focus of this work (similarly to [3, 1]) is on reducing the complexity of the self-attention component.

the complexity of which is $O(TS^2)$. Notably, the complexity of the proposed space-time mixing attention is also $O(TS^2)$. Following spatial-only attention, simple temporal averaging is performed on the class tokens $\mathbf{z}_{final} = \frac{1}{T}\sum_t \mathbf{z}_{t,cls}^{L-1}$ to obtain a single feature that is fed to the linear classifier.

**Recent work** by [3, 1] has focused on reducing the cost $O(T^2S^2)$ of the full space-time attention of Eq. 4. Bertasius et al. [3] proposed the factorised attention:

$$\tilde{\mathbf{y}}_{s,t}^l = \sum_{t'=0}^{T-1} \text{Softmax}\{(\mathbf{q}_{s,t}^l \cdot \mathbf{k}_{s,t'}^l)/\sqrt{d_h}\}\mathbf{v}_{s,t'}^l, \qquad \begin{Bmatrix} s = 0,\ldots,S-1 \\ t = 0,\ldots,T-1 \end{Bmatrix}, \qquad (6)$$
$$\mathbf{y}_{s,t}^l = \sum_{s'=0}^{S-1} \text{Softmax}\{\tilde{\mathbf{q}}_{s,t}^l \cdot \tilde{\mathbf{k}}_{s',t}^l)/\sqrt{d_h}\}\tilde{\mathbf{v}}_{s',t}^l,$$

where $\tilde{\mathbf{q}}_{s,t}^l, \tilde{\mathbf{k}}_{s',t}^l \tilde{\mathbf{v}}_{s',t}^l$ are new query, key and value vectors calculated from $\tilde{\mathbf{y}}_{s,t}^l$ [4]. The above model reduces complexity to $O(T^2S + TS^2)$. However, temporal attention is performed for a fixed spatial location which is ineffective when there is camera or object motion and there is spatial misalignment between frames.

The work of [1] is concurrent to ours and proposes the following approximation: $L_s$ Transformer layers perform spatial-only attention as in Eq. 5 (each with complexity $O(S^2)$). Following this, there are $L_t$ Transformer layers performing temporal-only attention on the class tokens $\mathbf{z}_t^{L_s}$. The complexity of the temporal-only attention is, in general, $O(T^2)$.

**Our model** aims to better approximate the full space-time self-attention (SA) of Eq. 4 while keeping complexity to $O(TS^2)$, i.e. inducing no further complexity to a spatial-only model.

To achieve this, we make a first approximation to perform full space-time attention but restricted to a local temporal window $[-t_w, t_w]$:

$$\mathbf{y}_{s,t}^l = \sum_{t'=t-t_w}^{t+t_w}\sum_{s'=0}^{S-1} \text{Softmax}\{(\mathbf{q}_{s,t}^l \cdot \mathbf{k}_{s',t'}^l)/\sqrt{d_h}\}\mathbf{v}_{s',t'}^l = \sum_{t'=t-t_w}^{t+t_w} \mathbf{V}_{t'}^l \mathbf{a}_{t'}^l, \ \{{}^{s=0,\ldots,S-1}_{t=0,\ldots,T-1}\} \quad (7)$$

where $\mathbf{V}_{t'}^l = [\mathbf{v}_{0,t'}^l; \mathbf{v}_{1,t'}^l; \ldots; \mathbf{v}_{S-1,t'}^l] \in \mathbb{R}^{d_h \times S}$ and $\mathbf{a}_{t'}^l = [a_{0,t'}^l, a_{1,t'}^l, \ldots, a_{S-1,t'}^l] \in \mathbb{R}^S$ is the vector with the corresponding attention weights. Eq. 7 shows that, for a single Transformer layer, $\mathbf{y}_{s,t}^l$ is a spatio-temporal combination of the visual tokens in the local window $[-t_w, t_w]$. It follows that, after $k$ Transformer layers, $\mathbf{y}_{s,t}^{l+k}$ will be a spatio-temporal combination of the visual tokens in the local window $[-kt_w, kt_w]$ which in turn conveniently allows to perform spatio-temporal attention over the whole clip. For example, for $t_w = 1$ and $k = 4$, the local window becomes $[-4, 4]$ which spans the whole video clip for the typical case $T = 8$.

The complexity of the local self-attention of Eq. 7 is $O(T(2t_w + 1)^2S^2)$. To reduce this even further, we make a second approximation on top of the first one as follows: the attention between spatial locations $s$ and $s'$ according to the model of Eq. 7 is:

$$\sum_{t'=t-t_w}^{t+t_w} \text{Softmax}\{(\mathbf{q}_{s,t}^l \cdot \mathbf{k}_{s',t'}^l)/\sqrt{d_h}\}\mathbf{v}_{s',t'}^l, \qquad (8)$$

i.e. it requires the calculation of $2t_w + 1$ attentions, one per temporal location over $[-t_w, t_w]$. Instead, we propose to calculate a single attention over $[-t_w, t_w]$ which can be achieved by $\mathbf{q}_{s,t}^l$ attending $\mathbf{k}_{s',-t_w:t_w}^l \triangleq [\mathbf{k}_{s',t-t_w}^l; \ldots; \mathbf{k}_{s',t+t_w}^l] \in \mathbb{R}^{(2t_w+1)d_h}$. Note that to match the dimensions of $\mathbf{q}_{s,t}^l$ and $\mathbf{k}_{s',-t_w:t_w}^l$ a further projection of $\mathbf{k}_{s',-t_w:t_w}^l$ to $\mathbb{R}^{d_h}$ is normally required which has complexity $O((2t_w + 1)d_h^2)$ and hence compromises the goal of an efficient implementation. To alleviate this we use the "shift trick" [45, 26] which allows to perform both zero-cost dimensionality reduction, space-time mixing and attention (between $\mathbf{q}_{s,t}^l$ and $\mathbf{k}_{s',-t_w:t_w}^l$) in $O(d_h)$. In particular, each $t' \in [-t_w, t_w]$ is assigned $d_h^{t'}$ channels from $d_h$ (i.e. $\sum_{t'} d_h^{t'} = d_h$). Let $\mathbf{k}_{s',t'}^l(d_h^{t'}) \in \mathbb{R}^{d_h^{t'}}$ denote the operator for

---

[4]More precisely, Eq. 6 holds for $h = 1$ heads. For $h > 1$, the different heads $\tilde{\mathbf{y}}_{s,t}^{l,h}$ are concatenated and projected to produce $\tilde{\mathbf{y}}_{s,t}^l$.

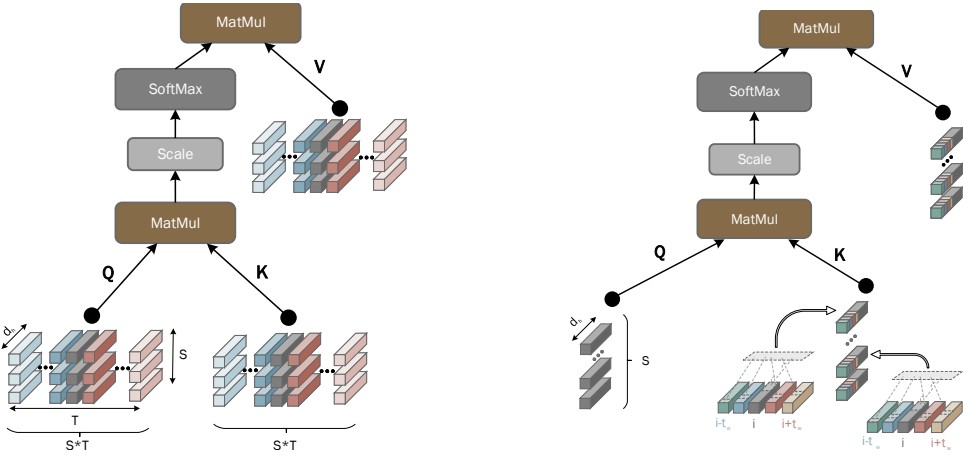

|  (a) Full space-time attention. | (b) Proposed space-time mixing attention. |

Figure 2: Detailed self-attention computation graph for (a) full space-time attention and (b) the proposed space-time mixing approximation. Notice that in our case only S tokens participate instead of ST. The temporal information is aggregated by indexing channels from adjacent frames. Tokens of identical colors share the same temporal index.

indexing the $d_h^{t'}$ channels from $\mathbf{k}_{s',t'}^l$. Then, a new key vector is constructed as:

$$\tilde{\mathbf{k}}_{s',-t_w:t_w}^l \triangleq [\mathbf{k}_{s',t-t_w}^l(d_h^{t-t_w}), \ldots, \mathbf{k}_{s',t+t_w}^l(d_h^{t+t_w})] \in \mathbb{R}^{d_h}. \tag{9}$$

Fig. 2 shows how the key vector $\tilde{\mathbf{k}}_{s',-t_w:t_w}^l$ is constructed. In a similar way, we also construct a new value vector $\tilde{\mathbf{v}}_{s',-t_w:t_w}^l$. Finally, the proposed approximation to the full space-time attention is given by:

$$\mathbf{y}_{s,t}^{l_s} = \sum_{s'=0}^{S-1} \text{Softmax}\{(\mathbf{q}_{s,t}^{l_s} \cdot \tilde{\mathbf{k}}_{s',-t_w:t_w}^l / \sqrt{d_h}\} \tilde{\mathbf{v}}_{s',-t_w:t_w}^l, \left\{ {s=0,\ldots,S-1 \atop t=0,\ldots,T-1} \right\}. \tag{10}$$

This has the complexity of a spatial-only attention ($O(TS^2)$) and hence it is more efficient than previously proposed video transformers [3, 1]. Our model also provides a better approximation to the full space-time attention and as shown by our results it significantly outperforms [3, 1].

**Temporal Attention aggregation:** The final set of the class tokens $\mathbf{z}_{t,cls}^{L-1}, 0 \leq t \leq L-1$ are used to generate the predictions. To this end, we propose to consider the following options: (a) simple temporal averaging $\mathbf{z}_{final} = \frac{1}{T} \sum_t \mathbf{z}_{t,cls}^{L-1}$ as in the case of our baseline. (b) An obvious limitation of temporal averaging is that the output is treated purely as an ensemble of per-frame features and, hence, completely ignores the temporal ordering between them. To address this, we propose to use a lightweight Temporal Attention (TA) mechanism that will attend to the $T$ classification tokens. In particular a $\mathbf{z}_{final}$ token attends the sequence $[\mathbf{z}_{0,cls}^{L-1}, \ldots, \mathbf{z}_{T-1,cls}^{L-1}]$ using a temporal Transformer layer and then fed as input to the classifier. This is akin to the (concurrent) work of [1] with the difference being that in our model we found that a single TA layer suffices whereas [1] uses $L_t$. A consequence of this is that the complexity of our layer is $O(T)$ vs $O(2(L_t-1)T^2 + T)$ of [1].

**Summary token:** As an alternative to TA, herein, we also propose a simple lightweight mechanism for information exchange between different frames at intermediate layers of the network. Given the set of tokens for each frame $t$, $\mathbf{Z}_t^{l-1} \in \mathbb{R}^{(S+1) \times d_h}$ (constructed by concatenating all tokens $\mathbf{z}_{s,t}^{l-1}, s = 0, \ldots, S$), we compute a new set of $R$ tokens $\mathbf{Z}_{r,t}^l = \phi(\mathbf{Z}_t^{l-1}) \in \mathbb{R}^{R \times d_h}$ which summarize the frame information and hence are named "Summary" tokens. These are then, appended to the visual tokens of all frames to calculate the keys and values so that the query vectors attend the original keys plus the Summary tokens. Herein, we explore the case that $\phi(.)$ performs simple spatial averaging $\mathbf{z}_{0,t}^l = \frac{1}{S} \sum_s \mathbf{z}_{s,t}^l$ over the tokens of each frame ($R = 1$ for this case). Note that, for $R = 1$, the extra cost that the Summary token induces is $O(TS)$.

**X-ViT:** We call the Video Transformer based on the proposed (a) space-time mixing attention and (b) lightweight global temporal attention (or summary token) as **X-ViT**.

# 4 Results

## 4.1 Experimental setup

**Datasets:** We train and evaluate the proposed models on the following datasets (all datasets are publicly available for research purposes):

*Kinetics-400 and 600*: The Kinetics [21] dataset consists of short clips (typically 10 sec long sampled from YouTube) labeled using 400 and 600 classes, respectively. Due to the removal of some videos from YouTube, the version of the dataset used in this paper consists of approximately 261K clips for Kinetics-400. Note, that these amounts are lower than the original version of the datasets and thus might represent a negative performance bias when compared with prior works.

*Something-Something-v2 (SSv2)*: The SSv2 [17] dataset consists of 220,487 short videos (of duration between 2 and 6 sec) that depict humans performing pre-defined basic actions with everyday objects. Because the objects and backgrounds in the videos are consistent across different action classes, this dataset tends to require stronger temporal modeling. Due to this, we conducted most of our ablation studies on SSv2 to better analyze the importance of the proposed components.

*Epic Kitchens-100 (Epic-100)*: is an egocentric large scale action recognition dataset consisting of more than 90,000 action segments spanning 100 hours of recordings in home environments, capturing daily activities [10]. The dataset is labeled using 97 verb classes and 300 noun classes. The evaluation results are reported using the standard action recognition protocol: the network predicts the "verb" and the "noun" using two heads. The predictions are then merged to construct an "action" which is used to report the accuracy.

**Training details:** All models, unless otherwise stated, were trained using the following scheduler and training procedure: specifically, our models were trained using SGD with momentum (0.9) and a cosine scheduler [28] (with linear warmup) for 35 epochs on SSv2, 50 on Epic-100 and 30 on Kinetics. The base learning rate, set at a batch size of 128, was 0.05 (0.03 for Kinetics). To prevent over-fitting we made use of the following augmentation techniques: random scaling ($0.9\times$ to $1.3\times$) and cropping, random flipping (with probability of 0.5; not for SSv2) and autoaugment [8]. In addition, for SSv2 and Epic-100, we also applied random erasing (probability=0.5, min. area=0.02, max. area=1/3, min. aspect=0.3) [52] and label smoothing ($\lambda = 0.3$) [34] while, for Kinetics, we used mixup [51] ($\alpha = 0.4$).

Table 1: Effect of local window size. To isolate its effect from that of temporal aggregation, the models were trained using temporal averaging. Note, that *(Bo.)* indicates that only features from the boundaries of the local window were used, ignoring the intermediate ones.

| Variant | Top-1 | Top-5 |
|---------|-------|-------|
| $t_w = 0$ | 45.2 | 71.4 |
| $t_w = 1$ | **62.5** | **87.8** |
| $t_w = 2$ | 60.5 | 86.4 |
| $t_w = 2$ (Bo.) | 60.4 | 86.2 |

The backbone models follow closely the ViT architecture of Dosovitskiy et al. [13]. Most experiments were performed using the ViT-B/16 variant ($L = 12$, $h = 12$, $d = 768$, $K = 16$), where $L$ represents the number of transformer layers, $h$ the number of heads, $d$ the embedding dimension and $K$ the patch size. We initialized our models from a pretrained ImageNet-21k [11] ViT model. The spatial positional encoding $\mathbf{p}_s$ was initialized from the pretrained 2D model and the temporal one, $\mathbf{p}_t$, with zeros so that it does not have a great impact on the tokens early on during training. The models were trained on 8 V100 GPUs using PyTorch [30].

**Testing details:** Unless otherwise stated, we used ViT-B/16 and $T = 8$ frames. We mostly used Temporal Attention (TA) for temporal aggregation. We report accuracy results for $1 \times 3$ views (1 temporal clip and 3 spatial crops) departing from the common approach of using up to $10 \times 3$ views [26, 16]. The $1 \times 3$ views setting was also used in Bertasius et al. [3]. To measure the variation between runs, we trained one of the 8–frame models 5 times. The results varied by $\pm 0.4\%$.

## 4.2 Ablation studies

Throughout this section, we study the effect of varying certain design choices and different components of our method. Because SSv2 tends to require a more fine-grained temporal modeling, unless otherwise specified, all results reported, in this section, are on the SSv2.

Table 2: Effect of: (a) proposed SA position, (b) temporal aggregation and number of Temporal Attention (TA) layers, (c) space-time mixing $qkv$ vectors and (d) amount of mixed channels on SSv2.

(a) Effect of applying the proposed SA to certain layers.

| Transform. layers | Top-1 | Top-5 |
|---|---|---|
| 1st half | 61.7 | 86.5 |
| 2nd half | 61.6 | 86.3 |
| Half (odd. pos) | 61.2 | 86.4 |
| All | **62.6** | **87.8** |

(b) Effect of number of TA layers. 0 corresponds to temporal averaging.

| #. TA layers | Top-1 | Top-5 |
|---|---|---|
| 0 (temp. avg.) | 62.4 | 87.8 |
| 1 | 64.4 | **89.3** |
| 2 | **64.5** | **89.3** |
| 3 | **64.5** | **89.3** |

(c) Effect of space-time mixing. x denotes the input token before $qkv$ projection. Query produces equivalent results with key and thus omitted.

| x | key | value | Top-1 | Top-5 |
|---|---|---|---|---|
| ✗ | ✗ | ✗ | 56.6 | 83.5 |
| ✓ | ✗ | ✗ | 63.1 | 88.8 |
| ✗ | ✓ | ✗ | 63.1 | 88.8 |
| ✗ | ✗ | ✓ | 62.5 | 88.6 |
| ✗ | ✓ | ✓ | **64.4** | **89.3** |

(d) Effect of amount of mixed channels. * uses temp. avg. aggregation.

| 0%* | 0% | 25% | 50% | 100% |
|---|---|---|---|---|
| 45.2 | 56.6 | 64.3 | **64.4** | 62.5 |

**Effect of local window size:** Table 1 shows the accuracy of our model by varying the local window size $[-t_w, t_w]$ used in the proposed space-time mixing attention. Firstly, we observe that the proposed model is significantly superior to our baseline ($t_w = 0$) which uses spatial-only attention. Secondly, a window of $t_w = 1$ produces the best results. This shows that more gradual increase of the effective window size that is attended is more beneficial compared to more aggressive ones, i.e. the case where $t_w = 2$. A performance degradation for the case $t_w = 2$ could be attributed to boundary effects (handled by filling with zeros) which are aggravated as $t_w$ increases. Based on these results, we chose to use $t_w = 1$ for the models reported hereafter. For short to medium long videos, it seems that $t_w = 1$ suffices as the temporal receptive field size increases as we advance in depth in the model allowing it to capture a larger effective temporal window. For the datasets used, as explained earlier, after a few transformer layers the whole clip is effectively covered. However, for significantly longer video sequences, larger window sizes may perform better.

**Effect of SA position:** We explored which layers should the proposed space-time mixing attention be applied to *within the network*. Specifically, we explored the following variants: Applying it to the first $L/2$ layers, to the last $L/2$ layers, to every odd indexed layer and, finally, to all layers. As the results from Table 2a show, the exact layers within the network that self-attention is applied to do not matter; what matters is the number of layers it is applied to. We attribute this result to the increased temporal receptive field and cross-frame interactions.

**Effect of temporal aggregation:** Herein, we compare the two methods used for temporal aggregation: simple temporal averaging [41] and the proposed Temporal Attention (TA) mechanism. Given that our model already incorporates temporal information through the proposed space-time attention, we also explored how many TA layers are needed. As shown in Table 2b, replacing temporal averaging with one TA layer improves the Top-1 accuracy from 62.5% to 64.4%. Increasing the number of layers further yields no additional benefits. In Table 2d, we also report the accuracy of spatial-only attention (0% mixing) plus TA aggregation. In the absence of the pro-

Table 3: Effect of number of tokens on SSv2.

| Variant | Top-1 | Top-5 |
|---|---|---|
| XViT-T/16 | 54.7 | 82.8 |
| XViT-S/32 | 57.0 | 84.6 |
| XViT-S/16 | 61.1 | 88.0 |
| XViT-B/32 | 60.5 | 87.4 |
| XViT-L/32 | 61.8 | 88.3 |
| XViT-B/16 | **64.4** | **89.3** |

posed space-time mixing attention, the TA layer alone is unable to compensate, scoring only 56.6%. In the same table, 45.2% is the accuracy of a model trained without the proposed local attention and TA layer (*i.e.* using a temporal pooling for aggregation). Overall, the results highlight the need of having both components in our final model. For the next two ablation studies, we used 1 TA layer.

**Effect of space-time mixing $qkv$ vectors:** Paramount to our work is the proposed space-time mixing attention of Eq. 10 which is implemented by constructing $\tilde{\mathbf{k}}^l_{s', -t_w:t_w}$ and $\tilde{\mathbf{v}}^l_{s', -t_w:t_w}$ efficiently via channel indexing (see Eq. 9). Space-time mixing though can be applied in several different ways in

Table 4: Comparison between TA and Summary token on SSv2 (left) and Kinetics-400 (right).

| Summary | TA | Top-1 | Top-5 |
|---|---|---|---|
| ✗ | ✗ | 62.4 | 87.8 |
| ✓ | ✗ | 63.7 | 88.9 |
| ✓ | ✓ | 63.4 | 88.9 |
| ✗ | ✓ | **64.4** | **89.3** |

| Summary | TA | Top-1 | Top-5 |
|---|---|---|---|
| ✗ | ✗ | 77.8 | 93.7 |
| ✓ | ✗ | **78.7** | **93.7** |
| ✓ | ✓ | 78.0 | 93.2 |
| ✗ | ✓ | 78.5 | **93.7** |

Table 5: Comparison with state-of-the-art on the Kinetics-400.

| Method | Top-1 | Top-5 | # Frames | Views | Params | FLOPs ($\times 10^9$) |
|---|---|---|---|---|---|---|
| bLVNet [14] | 73.5 | 91.2 | $24 \times 2$ | $3 \times 3$ | 25M | 840 |
| STM [19] | 73.7 | 91.6 | 16 | - | 24M | - |
| TEA [25] | 76.1 | 92.5 | 16 | $10 \times 3$ | 25.6M | 2,100 |
| TSM R50 [26] | 74.7 | - | 16 | $10 \times 3$ | 25.6M | 650 |
| I3D NL [44] | 77.7 | 93.3 | 128 | $10 \times 3$ | - | 10,800 |
| CorrNet-101 [40] | 79.2 | - | 32 | $10 \times 3$ | - | 6,700 |
| ip-CSN-152 [38] | 79.2 | 93.8 | 8 | $10 \times 3$ | - | 3,270 |
| LGD-3D R101 [31] | 79.4 | 94.4 | 16 | - | - | - |
| SlowFast 8×8 R101+NL [16] | 78.7 | 93.5 | 8 | $10 \times 3$ | - | 3,480 |
| SlowFast 16×8 R101+NL [16] | 79.8 | 93.9 | 16 | $10 \times 3$ | - | 7,020 |
| X3D-XXL [15] | 80.4 | 94.6 | - | $10 \times 3$ | 20.3M | 5,823 |
| TimeSformer-L [3] | **80.7** | 94.7 | 96 | $1 \times 3$ | 121M | 7,140 |
| ViViT-L/16x2 [1] | 80.6 | 94.7 | 32 | $4 \times 3$ | 312M | 17,352 |
| X-ViT (Ours) | 78.5 | 93.7 | 8 | $1 \times 3$ | 92M | 425 |
| X-ViT (Ours) | 79.4 | 93.9 | 8 | $2 \times 3$ | 92M | 850 |
| X-ViT (Ours) | 80.2 | 94.7 | 16 | $1 \times 3$ | 92M | 850 |
| X-ViT (Ours) | **80.7** | **94.7** | 16 | $2 \times 3$ | 92M | 1700 |

the model. For completeness, herein, we study the effect of applying space-time mixing to various combinations for the key, value and to the input token prior to $qkv$ projection. As shown in Table 2c, the combination corresponding to our model (*i.e.* space-time mixing applied to the key and value) significantly outperforms all other variants by up to 2%. This result is important as it confirms that our model, derived from the proposed approximation to the local space-time attention, gives the best results when compared to other non-well motivated variants.

**Effect of amount of space-time mixing:** We define as $\rho d_h$ the total number of channels coming from the adjacent frames in the local temporal window $[-t_w, t_w]$ (*i.e.* $\sum_{t'=-t_w, t\neq 0}^{t_w} d_h^{t'} = \rho d_h$) when constructing $\tilde{k}_{s', -t_w:t_w}^l$ (see Section 3). Herein, we study the effect of $\rho$ on the model's accuracy. As the results from Table 2d show, the optimal $\rho$ is between 25% and 50%. Increasing $\rho$ to 100% (*i.e.* all channels are coming from adjacent frames) unsurprisingly degrades the performance as it excludes the case $t' = t$ when performing the self-attention.

**Effect of Summary token:** Herein, we compare Temporal Attention with Summary token on SSv2 and Kinetics-400. We used both datasets for this case as they require different type of understanding: fine-grained temporal (SSv2) and spatial content (Kinetics-400). From Table 4, we conclude that the Summary token compares favorable on Kinetics-400 but not on SSv2 showing that it is more useful in terms of capturing spatial information. Since the improvement is small, we conclude that 1 TA layer is the best global attention-based mechanism for improving the accuracy of our method adding also negligible computational cost.

**Effect of number of input frames:** Herein we evaluate the impact of increasing the number of input frames $T$ from 8 to 16 and 32. We note that, for our method, this change results in a linear increase in complexity. As the results from Table 7 show, increasing the number of frames from 8 to 16 offers a 1.8% boost in Top-1 accuracy on SSv2. Moreover, increasing the number of frames to 32 improves the performance by a further 0.2%, offering diminishing returns. Similar behavior can be observed on Kinetics and Epic-100 in Tables 5 and 8.

Table 6: Comparison with state-of-the-art on the Kinetics-600 dataset. $T\times$ is the number of frames used by our method.

| Method | Top-1 | Top-5 | Views | FLOPs ($\times 10^9$) |
|---|---|---|---|---|
| AttentionNAS [43] | 79.8 | 94.4 | - | 1,034 |
| LGD-3D R101 [31] | 81.5 | 95.6 | $10 \times 3$ | - |
| SlowFast R101+NL [16] | 81.8 | 95.1 | $10 \times 3$ | 3,480 |
| X3D-XL [15] | 81.9 | 95.5 | $10 \times 3$ | 1,452 |
| TimeSformer-HR [3] | 82.4 | 96.0 | $1 \times 3$ | 5,110 |
| ViViT-L/16x2 [1] | 82.5 | 95.6 | $4 \times 3$ | 17,352 |
| X-ViT ($8\times$) (Ours) | 82.5 | 95.4 | $1 \times 3$ | 425 |
| X-ViT ($16\times$) (Ours) | **84.5** | **96.3** | $1 \times 3$ | 850 |

**Effect of number of tokens and different model sizes:** Herein, we vary the number of input tokens by changing the patch size $K$. As the results from Table 3 show, even when the number of tokens decreases significantly (*e.g.* ViT-B/32 or ViT-S/32) our approach is still able to produce results of satisfactory accuracy. The benefit of that is having a model which is significantly more efficient. Similar concusions can be observed when the model size (in terms of parameters and FLOPs) is varied. Our approach provides consistent results in all cases, showcasing its ability to scale well from tiny (XViT-T) to large (XViT-L) models.

**Latency and throughput considerations:** While the channel shifting operation used by the proposed space-time mixing attention is zero-FLOP, there is still a small cost associated with memory movement operations. In order to ascertain that the induced cost does not introduce noticeable performance degradation, we benchmarked a Vit-B/16 ($8\times$ frames) model using spatial-only attention and the proposed space-time mixing attention on 8 V100 GPUs and a batch size of 128. A model with spatial-only attention has a throughput of 312 fps while our model has 304 fps.

Table 7: Comparison with state-of-the-art on SSv2. * - pretrained on Kinetics 600

| Method | Top-1 | Top-5 | # Frames | Views | FLOPs ($\times 10^9$) |
|---|---|---|---|---|---|
| TRN [53] | 48.8 | 77.6 | 8 | - | - |
| SlowFast+multigrid [46] | 61.7 | - | - | $1 \times 3$ | - |
| TimeSformer-L [3] | 62.4 | - | 96 | $1 \times 3$ | 7,140 |
| TSM R50 [26] | 63.3 | 88.5 | 16 | $2 \times 3$ | - |
| STM [19] | 64.2 | 89.8 | 16 | - | - |
| MSNet [24] | 64.7 | 89.4 | 16 | - | - |
| TEA [25] | 65.1 | 89.9 | 16 | - | - |
| ViViT-L/16x2 [3] | 65.4 | 89.8 | 32 | $4 \times 3$ | 11,892 |
| X-ViT (Ours) | 64.4 | 89.3 | 8 | $1 \times 3$ | 425 |
| X-ViT (Ours) | 66.2 | 90.6 | 16 | $1 \times 3$ | 850 |
| X-ViT* (Ours) | **67.2** | **90.8** | 16 | $1 \times 3$ | 850 |
| X-ViT (Ours) | 66.4 | 90.7 | 32 | $1 \times 3$ | 1,270 |

## 4.3 Comparison to state-of-the-art

Our best model uses the proposed space-time mixing attention in all the Transformer layers and performs temporal aggregation using a single lightweight temporal transformer layer as described in Section 3. Unless otherwise specified, we report the results using the $1 \times 3$ configuration for the views (1 temporal and 3 spatial) for all datasets. Regarding related work on transformer-based video recognition [1, 3], we included their very best models trained on the same data as our models. For TimeSformer, this is typically the TimeSformer-L version. For ViVit, we used the 16x2 configuration, with factorized-encoding for Epic-100 and SS-v2 (as reported in Tables 6d and 6e in [1]) and the full version for Kinetics (as reported in Table 6a in [1]).

On **Kinetics-400**, we match the current state-of-the-art while having significantly lower computational complexity than the next two best recently proposed methods that also use Transformer-based architectures: $20\times$ fewer FLOPs than ViViT [1] and $8\times$ fewer than TimeSformer-L [3]. Note that both models from [1, 3] and ours were initialized from a ViT model pretrained on ImageNet-21k [11] and take as input frames at a resolution of $224 \times 224$px. Similar conclusions can be drawn from Table 6 which reports our results on Kinetics-600.

On **SSv2**, we match and surpass the current state-of-the-art, especially in terms of Top-5 accuracy (ours: 90.7% vs ViViT: 89.8% [1]) using models that are $14\times$ (16 frames) and $9\times$ (32 frames) faster.

Finally, we observe similar outcomes on **Epic-100** where we set a new state-of-the-art, showing large improvements especially for "Verb" accuracy, while again being more efficient.

## 5 Ethical considerations and broader impact

Current high-performing video recognition models tend to have high computational demands for both training and testing and, by extension, significant environmental costs. This is especially true for the transformer-based architectures. Our research introduces a novel approach that matches and surpasses the current state-of-the-art while being significantly more efficient thanks to the linear scaling of the complexity with respect to the number of frames. We hope such models will offer noticeable reduction in power consumption while setting at the same time a solid base for future research. We will release code and models to facilitate this. Moreover, and similarly to most data-driven systems, bias from the training data can potentially affect the fairness of the model. As such, we suggest to take this aspect into consideration when deploying the models into real-world scenarios.

Table 8: Comparison with state-of-the-art on Epic-100. $T\times$ is the #frames used by our method. Results for other methods are taken from [1].

| Method | Action | Verb | Noun |
|---|---|---|---|
| TSN [41] | 33.2 | 60.2 | 46.0 |
| TRN [53] | 35.3 | 65.9 | 45.4 |
| TBN [22] | 36.7 | 66.0 | 47.2 |
| TSM [22] | 38.3 | 67.9 | 49.0 |
| SlowFast [16] | 38.5 | 65.6 | 50.0 |
| ViViT-L/16x2 [1] | 44.0 | 66.4 | **56.8** |
| X-ViT ($8\times$) (Ours) | 41.5 | 66.7 | 53.3 |
| X-ViT ($16\times$) (Ours) | **44.3** | **68.7** | 56.4 |

## 6 Conclusions

We presented a novel approximation to the full space-time attention that is amenable to an efficient implementation and applied it to video recognition. Our approximation has the same computational cost as spatial-only attention yet the resulting video Transformer model was shown to be significantly more efficient than recently proposed Video Transformers [3, 1]. By no means this paper proposes a complete solution to video recognition using video Transformers. Future efforts could include combining our approaches with other architectures than the standard ViT, removing the dependency on pre-trained models and applying the model to other video-related tasks like detection and segmentation. Finally, further research is required for deploying our models on low power/resource devices.

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
