# Supplementary material for Space-time Mixing Attention for Video Transformer

**Adrian Bulat**
Samsung AI Cambridge
adrian@adrianbulat.com

**Juan-Manuel Perez-Rua**
Samsung AI Cambridge
j.perez-rua@samsung.com

**Swathikiran Sudhakaran**
Samsung AI Cambridge
swathikir.s@samsung.com

**Brais Martinez**
Samsung AI Cambridge
brais.a@samsung.com

**Georgios Tzimiropoulos**
Samsung AI Cambridge
Queen Mary University of London
g.tzimiropoulos@qmul.ac.uk

## A  Appendix

**Effect of the number of crops at test time:** Throughout this work, at test time, we report results using 1 temporal and 3 spatial crops (*i.e.* $1 \times 3$). This is noticeable different from the current practice of using up to $10 \times 3$ crops [4, 1].

To showcase the behavior of our method, herein, we test the effect of increasing the number of crops as measured on Kinetics-400. As the results from Fig. 1 show, increasing the number of crops beyond two temporal views (*i.e.* $2 \times 3$), yields no additional gains. Our findings align with the ones recently reported in Bertasius et al. [2].

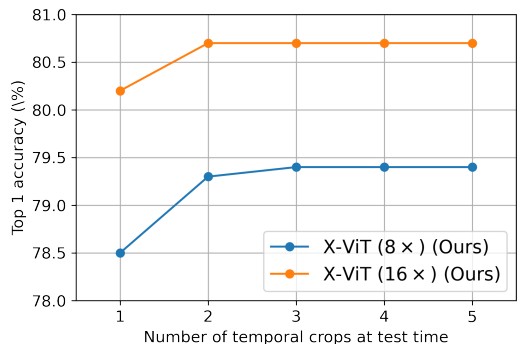

Figure 1: Effect of number of temporal crops at test time as measured on Kinetics-400 in terms of Top-1 accuracy. For each temporal crop, 3 spatial clips are sampled, for a total of $t_{crops} \times 3$ clips. Notice that beyond $t_{crops} = 2$ no additional accuracy gains are observed.

**Inference speed:** In the main paper, where available, we detailed the computational complexity of each model as measured in FLOPs. In Table 1, we show the complexity as measure in videos per second for a few selected models. The test is performed on 8 V100 GPUs.

**Comparison with TSM [6]:** To emphasise and experimentally validate the conceptual differences between TSM and the proposed X-ViT method (both sharing the use of the "shift trick" [8]"), we applied the method described in TSM to the ViT architecture. A straightforward application of the the shift trick to ViT, which from now on we call TSM-ViT, can be described by the following Equations:

35th Conference on Neural Information Processing Systems (NeurIPS 2021).

Table 1: Speed comparison in videos/sec. Where possible, the original, open-sourced implementations provided by the authors were used.

| Method | images/sec |
|---|---|
| bLVNet [3] | 7.5 |
| TEA [5] | 2.4 |
| i3D-NL ResNet101 [7] | 0.37 |
| SlowFast (8frames) [4] | 0.69 |
| TimeSformer [2] | 0.69 |
| ViViT-L [1] | 0.66 |
| X-ViT (Ours) | 12 |

$$\mathbf{Z}^l = \text{SHIFT}(\mathbf{Z}^l), \quad (1)$$

$$\mathbf{Y}^l = \text{MSA}(\text{LN}(\mathbf{Z}^{l-1})) + \mathbf{Z}^{l-1}, \quad (2)$$

$$\mathbf{Z}^l = \text{MLP}(\text{LN}(\mathbf{Y}^l)) + \mathbf{Y}^l. \quad (3)$$

Note that the differences between TSM-ViT and our model (XViT) are:

1. 1. TSM-ViT does not perform an approximation to the full space-time attention as XViT does.

2. TSM performs simple temporal average pooling for temporal aggregation. Instead we propose two new forms of aggregation: Temporal Attention aggregation and Summary Token.

The results of Table 2 clearly show that the two approaches are different. Moreover, we see that the proposed temporal attention aggregation is much more effective than the simple temporal pooling proposed by the original TSM paper.

Table 2: Comparison between XViT and "TSM-ViT" on the SS-v2 dataset in terms of Top-1 accuracy. ViT-B/16 was used for all models.

| Method | Top-1 |
|---|---|
| TSM-ViT (with temporal aggregation as proposed in TSM) | 60.8% |
| XViT (with temporal aggregation as proposed in TSM) | 62.5% |
| TSM-ViT (with Temporal Attention aggregation) | 63.1% |
| XViT (with Temporal Attention aggregation) | **64.4%** |