# OpenReview forum: "Space-time Mixing Attention for Video Transformer"
_NeurIPS.cc/2021/Conference — NeurIPS 2021 Poster_

### Official Review · Reviewer_qij4 · 2021-07-06

**Rating:** 6
**Confidence:** 3

**Summary:**

In this paper, the authors focus on reducing the computation cost caused by the space-time attention modeling in the Transformer framework for video recognition. Specifically, the proposal includes 1) a space-time mixing attention module that mixes channels from neighboring frames within a local window as key-value tokens, and 2) a temporal attention layer at the end of the model to aggregate global temporal information. The authors provide promising results on Kinetics-400 and SSv2 in the paper.

**Limitations And Societal Impact:**

It is good that the authors have discussed the ethical consideration and broader impact of their model in terms of environmental costs and the bias of data-driven systems. We suggest the authors provide specific cases and more details of the mentioned fairness of their model (in Line 314-315).

**Main Review:**

**Originality:** The major proposal is the space-time mixing attention module. The authors introduce the “channel shift trick [42]” to the space-time attention module to aggregate both spatial and temporal features while reducing computation complexity. They have shown promising results (Table 5-6) to support their claims. I think the idea is novel and clearly differs from previous works (TimeSformer [3] and ViViT[3]).

**Quality:** The submission is technically sound. The authors have provided extensive experiments and ablation studies to support their major claims. The weaknesses are list as below.
1. The authors should provide the typical experimental values of T, S, and d (Line 98-101), otherwise, it may be hard to figure out the dominant computation cost of a full model. For example, when $2d > TS$, the complexity of MLP is larger than MSA.
2. I’m not sure about the complexity mentioned in Line 129. In my understanding, in the local temporal window of Eq. 7, there are $(2t+1)S$ tokens, and I think the complexity should be $O(T(2t+1)^2S^2) \sim O(Tt^2S^2)$, rather than $O((2t+1)TS^2)$.
3. In my understanding, the space-temporal mixing attention mainly aggregates temporal information by aggregating a subset of channels from neighboring frames to constitute the features of a new key-value token (Line 137-140). I think another intuitive option is reducing channels of each frame (e.g., reduce to 1/t channels) and then concatenate them together as the final key-value tokens. Such a design can aggregate temporal information while also avoiding introducing additional overhead. We suggest the authors clarify their differences and prove the superiority of their proposal over the abovementioned option.
4. I think the results of Table 2 (b) look weird as the results of three different models are totally the same. We suggest the authors carefully check the results and report the error bar of the results.
5. The authors have highlighted the computation complexity and cost multiple times in the paper. Therefore, I think it is important to provide latency (i.e., runtime), the number of input frames, the number of tokens for all the models in Table 5 and Table 6 to support their claims.


**Clarity:** I think this paper is well organized, however, some details of the model are not clear enough.
1. it’s hard for me to totally understand the summary token (Line 159-166). We suggest the authors provide illustrations of the temporal attention aggregation and summary token to demonstrate their functions and differences.
2. In Table 2 (d), * (44.2) indicates temporal average aggregation, however, it is reported as 55.6% in Line 241. The authors should check it and clarify it.

**Significance:** This paper proposes a novel way to reduce computation cost for spatial-temporal attention modeling. The authors have shown promising FLOPs and accuracy in standard benchmarks. Latency and more details of Table 5 and Table 6 are expected in the final version. Overall, I think it is useful and helpful to others to follow similar ideas to build transformer networks for video tasks.

**Time Spent Reviewing:**

4 hours

---

> ### Author Response · Authors · 2021-08-10
> **Response to reviewer 4 (qij4)**
>
> **R4.1:** On providing the typical experimental values of T, S and d.
>
> **A4.1:** Thank you for your suggestion. Typically  S=196, T=\{8,16,32\} and d=768 (for a ViT-B backbone). You are right that the MLP complexity is by no means negligible, however the focus of our paper (similar to [1,3]) is on reducing the complexity of the self-attention part. We will mention this in the revised paper.
>
> **R4.2:** On the complexity from L129).
>
> **A4.2:** You're right a square is missing from there. Thank you for spotting this!
>
> **R4.3:** On showing that the proposed methods is better than the suggested variation: reducing the channels of each frame to 1/t and the concatenating them together.
>
> **A4.3:** Thank you for this suggestion. We tested your idea (at least the way we interpreted it!) on the SS-v2 dataset, taking the freedom to explore a few variations around it such as how the reduction is performed. The best variant scores 60.4\% using 8 frames. This is better than the baseline without any temporal modelling (except for the last layer) that scores 55.6\% (see Table 2(c))  but worse than our  best result that scores 63.4\% (now 64.4\% after tuning the learning rate further). We believe that looking at all frames at once in such manner while decreasing the number of channels per frame within the concatenated tensor causes an information bottleneck which is exacerbated as the number of input frames increases since this will make 1/t smaller. We will include these extra results in the paper.
>
> **R4.4:** On the results from Table 2b.
>
> **A4.4:** We verified the results, running each experiment 3 times. As a result, we can confirm that they are correct. The table shows that using more than 1 temporal aggregation layer is not helpful on the datasets tested.
>
> **R4.5:** On reporting run-time performance.
>
> **A4.5:** Thank you for your suggestion. While generally the number of FLOPs is a good proxy metric, we will include the timings for all the methods that offer a public implementation. We are using the metric proposed by R3(fqvk) that is to measure the number of videos processed per second. The number of frames and clips required to produce the prediction for a video follows the setting described by each method.  We list bellow a few of these results:
>
> bLVNet: 7.5videos/sec; TEA: 2.4videos/sec; i3D-NL ResNet101: 0.37videos/sec; SlowFast (8f): 0.69videos/sec; TimeSformer: 0.69videos/sec; ViVit: 0.66videos/sec; XViT (Ours): 12 videos/sec;
>
> As it can be observed, our method is significantly faster, especially when compared against the concurrent work of [1,3] that also uses transformers.
>
> **R4.7:** On adding illustrations for L159-166.
>
> **A4.7:** Thank you for your suggestion, we will add a figure explaining this in the supplementary material.
>
> **R4.8:**  On using temporal avg. results 44.2\% vs 55.6\%.
>
> **A4.8:** The 44.2\% result refers to a model trained without the proposed local attention and the temporal attention layer of L147-156. The final prediction in this case is produced using a "temporal aggregation layer" - i.e. a pooling operation across time. In comparison, the model scoring 55.6\% replaces the pooling operation across time with the temporal attention layer described in L147-156. We will make this clearer in the updated manuscript.

---

> > ### Comment · Reviewer_qij4 · 2021-08-23
> > **After checking Review 1's comments**
> >
> > Thank authors for the response.
> >
> > 1. I think some of my initial concerns are addressed (most are missed details/errors). However, as for the 1st issue listed in Clarity, I actually expected the authors could provide clarification about the summary token (one of the proposed contributions), rather than a promise in the final version.
> > 2. For reference, I have carefully read the review and rebuttal from Reviewer 1, who has higher confidence. I agree with the following concerns.
> >     - Experiments on different model sizes. After checking several typical works in this field, e.g., ip-CSN[36], LGD-3D[29], SlowFast[15], X3D[14], and ViT[12], I agree this kind of experiment is a classical experimental setting and is needed.
> >     - Novelty. From the rebuttal of A1.2, it seems that the key difference between X-ViT and TSM-ViT is the step of using the ‘shift trick’ (i.e., on global window VS. on local window). The only insight (not experiment result) the authors highlighted is the “mathematically approximation” but without rigid verification of assumption and proof.
> >
> >
> > Therefore, I tend to lower my rating.
> > Any discussions or more rebuttal are welcomed.

---

> > > ### Author Response · Authors · 2021-08-24
> > > **Response addressing the remaining concerns**
> > >
> > > Thank you for the additional feedback. We are happy that some of your concerns were addressed. We hope that we successfully address the remaining ones below.
> > >
> > > ### **1.** On further clarification regarding the summary token.
> > >
> > > The summary token acts as global representation of each frame computed by aggregating all the tokens within a frame. To construct them, an aggregation function is used which, in its simplest instantiation, takes the form of an average-pooling function. So, if the input to the MSA at layer $l$ and frame $t$ is noted as $Z_t^{l-1}$, then we apply a function $\phi$ (say avg-pool) to obtain a single vector/token $\phi(Z_t^{l-1})=Z_{t,r}^{l-1}$.
> > >
> > > For each token in a given frame $t$, the attention is computed over the set formed by concatenating all $N$ spatial tokens belonging to that frame with the $T$ global summaries computed from all frames, where $T$ denotes the number of input frames. Thus, the total number of tokens attended to are $N + T$ (usually a negligible increase over $N$ since $N >> T$).
> > >
> > > In effect, this introduces a direct interaction between global and local information, where the global information consists of a single token summarizing the content of the rest of the frames.
> > >
> > > Note that the "Temporal Attention aggregation" (L147-L156) is adding global temporal context at the end of the network, and "Summary token" is adding it interleaved.
> > >
> > > ### **2.1.** On experiments with different model sizes.
> > >
> > > We agree that providing results with different model sizes is useful so we trained more models and included their accuracy in the updated version. Additional results that show the accuracy across different model sizes (Tiny, Small, Base, and Large) can be seen below (please note that some of them were already present in the initial version of the manuscript in Table 4.).
> > >
> > > Note that we report results for 4 different model sizes (Tiny, Small, Base, and Large) which is already more sizes than Timesformer [1] (they report S, B, L) and ViVit [3] (they report B, L, H). Moreover, compared to Timesformer [1] our S-model is just 2\% worse than our B-model whereas in [1] the drop is 5\%. Finally, we also report results with different patch sizes.
> > >
> > > | Method    | Top-1 | Top-5 |
> > > |-----------|-------|-------|
> > > | XViT-T/16 | 54.7 | 82.8 |
> > > | XViT-S/32 | 57.0 | 84.6 |
> > > | XViT-S/16 | 61.1 | 88.0 |
> > > | XViT-B/32  |59.8 | 87.4  |
> > > | XViT-B/16   | 63.4 | 89.6  |
> > > | XViT-L/32   | 61.0 | 88.3 |
> > > | XViT-L/16 | 64.1 | 89.9 |
> > > *Table 1. Accuracy of XViT for different model sizes and patch sizes.*
> > >
> > > ### **2.2.** On novelty and similarities with TSM-ViT .
> > >
> > > Thank you for providing this comment as more clarifications were required.
> > >
> > > A straightforward application of the the shift trick to ViT, which from now on we will call TSM-ViT, can be described by the following Equations (note that this has not been described in some other paper before):
> > >
> > > $\mathbf{Z}^{l} = \texttt{SHIFT}(\mathbf{Z}^{l}) $
> > >
> > > $\mathbf{Y}^{l}  =  \textrm{MSA}(\textrm{LN}(\mathbf{Z}^{l-1})) + \mathbf{Z}^{l-1},$
> > >
> > > $\mathbf{Z}^{l+1}  = \textrm{MLP}(\textrm{LN}(\mathbf{Y}^{l})) + \mathbf{Y}^{l}. $
> > >
> > > Note that the differences between TSM-ViT and our model (XViT) are:
> > > 1. TSM-ViT does not perform a mathematically sound approximation to the full space-time attention as XViT does (described in our paper in **L118-L146** and **Eqs. (7)--(10)**).
> > > 2. TSM performs simple temporal average pooling for temporal aggregation. Instead we propose two new forms of aggregation: "Temporal Attention aggregation" (see **L147-156**) and "Summary Token" (see **L157-166**).
> > >
> > > The above model **was already evaluated in our paper in Table 2(c) second row**. For clarity, we report its accuracy in the following table together with XViT. Note that the first row in the table: *TSM-ViT (with final aggregation as proposed in TSM)* was not provided in the original submission.
> > >
> > > | Method    | Top-1 |
> > > |-----------|-------|
> > > | TSM-ViT (with final aggregation as proposed in TSM) | 59.8\% |
> > > | X-ViT (with final aggregation as proposed in TSM) | 61.6\% |
> > > | TSM-ViT (with final aggregation as proposed in L147-156) | 62.1\% |
> > > | X-ViT  (with final aggregation as proposed in L147-156) | 63.4\% |
> > > *Table 2. Comparison between XViT and "TSM-ViT". ViT-B/16 was used for all models.*
> > >
> > > We believe that these results clearly show that these two approaches are different. This is to be expected as XViT is the only one that performs a mathematically sound approximation to the full space-time attention. Moreover, we see that the proposed temporal attention aggregation is much more effective than the simple temporal pooling proposed by the original TSM paper.
> > >
> > >
> > > Should additional clarification be required, we would be more than happy to provide it. We are looking forward to your reply.

---

> > > > ### Comment · Reviewer_qij4 · 2021-08-25
> > > > **Thanks**
> > > >
> > > > Thanks for the response.
> > > > 1. Experiments. I think the improvements have been well verified.
> > > > 2. Novelty. The authors highlighted one of the biggest differences between X-ViT and TSM-ViT as "mathematically sound approximation". Actually, I think it is just a "statement/expression" rather than a "sound mathematical proof for the correctness of such an approximation". I suggest the authors do not overclaim it and focus on the true difference.
> > > >
> > > > I agree that the novelty of this paper is not significant.
> > > > However, considering the improvements and possible insights, I think the overall quality is marginally above the threshold. Therefore, I decided to keep my original rating.
> > > >
> > > > Thanks!

---

> > > > > ### Author Response · Authors · 2021-08-25
> > > > > **Thank you for you promt response and feedback**
> > > > >
> > > > > Thank you for your prompt response and feedback — we appreciate it.
> > > > >
> > > > > We would like to clarify that what we meant is that XViT does perform an approximation to full space time attention as derived in L118-L146. It’s unclear what TSM-XViT does and as shown it’s outperformed by XViT. Adding to this and by providing the experimental results of Table 2 (in our recent response to you), we also hope that we addressed your previous comments related to “only insight (not experiment result)” and then later “without rigid verification of assumption”.
> > > > > Our verification is only experimental though and we agree with you that we don’t provide a “proof for the correctness of such an approximation". This is similar in spirit to approximations provided by other works such Timesformer [1] and ViViT [3], which are also verified experimentally only.
> > > > >
> > > > > Thank you again for taking the time to review our responses

---

### Official Review · Reviewer_fqvk · 2021-07-14

**Rating:** 6
**Confidence:** 4

**Summary:**

The authors propose a more efficient self-attention operation, specifically for video models, such that the compute requirements increase linearly with the length of the video. The author's contributions consist of two main things: 1) Self-attention along the temporal axis is limited to a smaller window instead of the whole sequence (with multiple transformer layers, the temporal "receptive field" grows to cover the whole video) and 2) The authors efficiently perform "space-time mixing" by constructing the keys and values for attention by concatenating features from different frames. This operation requires 0 floating point operations, but is still able to combine temporal information from adjacent frames effectively.

The authors show good experimental results. They achieve similar results to ViViT [1] and TimeSFormer [3] whilst using far fewer GFLOPs, on Kinetics, SSv2 and Epic Kitchens. The paper is also well written in general.

**Limitations And Societal Impact:**

Adequately addressed in the main paper.

**Main Review:**

The method appears fairly straightforward to implement, is efficient in terms of GFLOPs and achieves comparable results to ViViT and TimeSFormer whilst using less GFLOPs. However, I do think the novelty is quite limited:

- The idea of restricting self-attention to more localised patterns has been explored before in a number of papers, ie [A, B, C, D] among others. The authors should discuss these papers more.
- The author's proposed method for efficient "space-time mixing" is very similar to that of TSM [42]. Moreover, although I am not aware of other transformer papers that do efficient "space-time mixing" in exactly the same way, it does seem quite similar in spirit to Model 4 of Vivit [1], and a discussion of this is completely missing in the paper.
- On a related note, the authors often refer to recent video-transformer papers [1, 3]. However, both of these papers propose multiple different models, and it's not clear which models in the paper the authors are referring to. For example in Figure 1, the attention pattern for TimeSFormer is their "Divided Space-Time Attention", and is also Model 3 of Vivit. On Page 5, when the authors are comparing their "Temporal Attention aggregation" to Vivit, this is referring only to Model 2 of [1]. These should be clarified in the revision.



In the experiments section, the authors state they match state-of-the-art results while being "significantly faster". Actually, the authors only report GFLOPs, so it is not correct to say that they are "faster". Rather, the proposed model uses less floating point operations, and so the authors should use more precise language here. (The same applies to the discussion of results in the supplementary) A further point here is that the author's "shift trick" does not require any FLOPs, but does increase the runtime. Note that TSM [42] also shows that the impact of the "shift trick" on runtime is architecture-specific with many caveats.

When the authors reported runtime on Page 8, it is reported with "frames/second" as the unit. A better unit would be "video clips / second" as that is independent of the number of frames used (as this is a model hyperparamter) and thus more comparable across different methods. Am I correct that as the authors process 8 frames per video, 312 frames/second = 39 video clips / second?

Table 4 would also be more clear if the number of tokens and/or number of frames processed by the model were included as separate columns.



Minor points

- Table 5: Timesformer and Vivit have the same reference.
- Line 155 to 156: [1] also shows that L_t = 1 works well, and this is identical to the author's temporal aggregation approach.



References

[A] N Parmar et al. Image Transformer. ICML 2018

[B] R Child et al. Generating long sequences with sparse transformers. ICML 2019

[C] I Betalgy et al. Longformer: The Long-Document Transformer (the authors cited this, but did not discuss it in context)

[D] M Zaheer et al. Big Bird: Transformers for Longer Sequences. NeurIPS 2020


**Time Spent Reviewing:**

5

---

> ### Author Response · Authors · 2021-08-10
> **Response to reviewer 3 (fqvk)**
>
> **R3.1:** On accuracy/efficiency of our method.
>
> **A3.1:** Thank you for recognizing that our method is much more efficient than TimeSformer and ViViT will being similarly accurate. By tweaking the learning rate slightly we managed to improve the results even further, now outperforming TimeSformer and ViViT using exactly the same setting as before. For example, on SS-v2, or results improved from 63.4\% (8 frames) to 64.4\% (8 frames) and from 65.2\% (16 frames) to 66.2\% (16 frames). Pre-training on Kinetics can further boost this to 67.2\% (16 frames).
>
>
> **R3.2:** On novelty and relation to [A,B,C,D].
>
> **A3.2:** Thank you for providing these papers. We already mention [C] which is probably the paper mostly related to our work. Our work is the first method to propose such formulation for space-time attention and video recognition which is the main focus of our paper. Another key differentiation is that our approach goes one step further (compared to the ones mentioned) by incorporating space-time mixing which avoids inducing any additional FLOPS and keeps the complexity linear with respect to the time dimension. We will also add a brief discussion of the remaining provided papers.
>
> **R3.3:** On similarity with TSM.
>
> **A3.3:** Our paper is not Transformer plus TSM or (more precisely) Transformer plus "shift trick" (as the zero flop shift trick was originally proposed in [42]). We would like to make 3 points here:
> 1. Firstly, by applying the "shift trick", not as described in our method, but let's say to the feature tensor prior to the application of the self-attention layer (2nd row of Table 2(c)) does not result in a mathematically sound approximation to space-time attention. In contrast, we propose a valid approximation to space-time attention as outlined in Eq. 10 and Fig. 1 (d). This is derived by firstly making the temporal local approximation of Eq. 7 and then appropriately applying the "shift trick" to the key and values in order to obtain an efficient implementation of Eq. 8 (the whole process is described in L120-145).
> 2. Secondly, we would like to draw your attention again to Table 2(c). The table clearly shows that is absolutely essential to use the proposed method, based on a mathematically sound approximation to space-time attention, in order to achieve the highest possible accuracy. For example, simply mixing the feature tensor as described above is more than 1\% worse (2nd row of Table 2(c)). Other ways of applying the space-time mixing give even worse results.
> 3.Finally, another point that we want to make is that we show that our model in terms of joint *space-time local* attention is significantly more accurate than factorized versions of self-attention ([1, 3]) which might be ineffective when there is camera or object motion and there is spatial misalignment  between frames.
>
> **R3.4:** On similarity with Model 4 of ViVit.
>
> **A3.4:** Our approach and Model 4 of ViViT are *unrelated*. As the authors of ViViT state, Model 4 is "similar in spirit to Model 3", but instead of factorizing over space and time separately (as in Model 3), they factorize the multi-head dot-product operation instead. Our approach does not perform any type of factorization (rather efficient joint space-time attention) and hence is unrelated to both Models 3 and 4 from ViVit. Furthermore, our approach matches or surpasses the accuracy of all of these variants while being more efficient in terms of number of FLOPs by an order of magnitude. These results clearly show the superiority of our formulation reinforcing the fact that they are not alike. We will make the differences very clear in the revised manuscript of our paper.
>
> **R3.5:** On similarity with other video transformers.
>
> **A3.5:** Thank you for providing these comments. You are right about Timesformer being equivalent to Model 3 from ViVit in Fig. 1. We will add this to Fig. 1. Moreover, in page 5, in lines 154-156, where the connection to ViVit is explained, we will make clear that this refers to Model 2.
>
> In the results section, we include the very best models from [1,3] trained on the same data as our models. For TimeSformer, this is typically the TimeSformer-L version. For ViVit, we use the 16x2 configuration, with factorized-encoding for Epic and SS-v2 (as reported in Tables 6d and 6e in [1]) and the full version for Kinetics (as reported in Table 6a in [1]). We will clarify this in the text and the tables to avoid any confusion. In general, we will make very clear which models from previous work we are referring to.
>
> **R3.6:** On significantly faster vs using less floating points.
>
> **A3.6:** Thank you, we will change the text to reflect that our model has significantly lower computational complexity with respect to the number of FLOPs.
>
> **R3.7:** On the impact of the "shift trick" on actual speed.
>
> **A3.7:** Indeed, there is a small impact on the runtime speed introduced by the shift trick, however this is very small. As you rightly notice, we report and discuss this in L279-284. There, it can be seen that the impact is minimal. The throughput decreases from 312 frames/sec to 304 frames/sec (With the shift trick).
>
> **R3.8:** On reporting runtime in terms of video clips/second.
>
> **A3.8:** Thank you for you suggestion. Your conversion it correct. We will report this metric too.
>
> **R3.9:** Reporting number of tokens and frames in a separate column.
>
> **A3.9:** Thank you for your suggestion. We created a new column with this data in the updated manuscript.
>
> **R3.10:** On  Timesformer and ViVit having the same reference in Table 4.
>
> **A3.10:** Thank you, fixed!
>
> **R3.11:** On L=1 working well for [1] too.
>
> **A3.11:** Yes but L=4 works better, and we simply mention what they used to report results for their paper. Also, it's unclear what kind of performance boost they get by using L=4 on other datasets like SS-v2 and Epic where the temporal dynamics are more important than Kinetics. For our model, we did not observe *any improvement* by using L>1 even on SS-v2 (see Table 2 (b)).

---

> > ### Comment · Reviewer_fqvk · 2021-08-28
> > **Update after rebuttal**
> >
> > Thank you for the rebuttal. It addresses my concerns (and I think the points raised by the other reviewers) quite well.
> > As pointed out by Reviewer qij4, I would not call the difference between X-ViT and TSM-ViT a "mathematically sound approximation", but the authors have described in the rebuttal how their method differs. I encourage the authors to update the paper accordingly.
> > Although I think the method is not particularly novel, the approach is fairly simple and effective, and the authors have ablated their approach thoroughly. As a result, i have upgraded my rating to lean towards acceptance.

---

### Official Review · Reviewer_kt96 · 2021-07-16

**Rating:** 6
**Confidence:** 3

**Summary:**

In this paper, the authors proposed a Video Transformer model the complexity of which scales linearly (instead of quadratic) with the number of frames in the video sequence. The key ideas are 1) restricting time attention to a local temporal window, and 2) using efficient space-time mixing to attend jointly spatial and temporal locations. They empirically verified that the proposed method outperforms existing state-of-the-art models in popular action recognition datasets.

**Limitations And Societal Impact:**

As mentioned in the main review #2, the proposed approach was verified only on a subclass of video datasets (short videos with homogeneous scenes, focusing on action recognition) but it was slightly over-claimed.


**Main Review:**

1. The paper is well-written overall, with full details of the model in Section 3 and intuitive demonstration in Figure 1.

2. It looks like t_w, the additional hyper-parameter of local window size, is very important and its optimal value may significantly vary dataset by dataset. Although the authors included an empirical evidence on Table 1, I still believe that this may need to be tuned for other datasets differently. Especially, when the dataset contains videos that are relative longer, and thus containing less homogeneous scenes, the optimal value for t_w may be significantly larger than 1. I guess the optimal value was 1 since SSv2 contains mostly homogeneous and short videos. This interpretation is aligned with the main idea of this work that attention only within a short temporal window is enough, as in Figure 1 (d). It is okay to limit the scope of this work to this kinds of short videos only and action recognition (as opposed to general, topical video classification for longer videos like YouTube 8M or TVR), but the scope still should be mentioned clearly.

3. Other than the point in #2, the experiment was conducted clearly, and shows impressive results on multiple datasets. Ablation studies were also conducted nicely.


**Time Spent Reviewing:**

3 hours

---

> ### Author Response · Authors · 2021-08-10
> **Response to reviewer 2 (kt96)**
>
> Thank you for this comment, this is a very good point which we will make sure to discuss and clarify in our paper. We agree that, for significantly longer video sequences, larger window sizes may perform better. Unfortunately training models for processing longer videos (e.g. $\geq 32$ frames) has proven challenging for our resources so we restricted ourselves to the standard action recognition benchmarks that most papers report results on. As you also pointed out, we would like to note that for short to medium sized videos, $t_w=1$ likely suffices as the temporal receptive field size will increase as we advance in depth in the model allowing us to capture a larger effective temporal window. For the datasets we used, as explained in L124-127 after a few transformer layers the whole clip is effectively covered.

---

> > ### Comment · Reviewer_kt96 · 2021-08-27
> > **Reply**
> >
> > Thanks for your comments and clarification.
> > I am keeping my original rating.

---

### Official Review · Reviewer_C6rH · 2021-07-18

**Rating:** 4
**Confidence:** 5

**Summary:**

This paper presents a video Transformer model for video recognition. It limits the temporal window to the local area and proposes efficient space-time mixing to reduce computational costs. On Kinetics, Something-Something v2, and Epic Kitchens datasets,  the proposed video transformer model is more effective than other transformer-based models yet has the same computational cost as spatial-only attention models.

**Limitations And Societal Impact:**

Yes.

**Main Review:**

For originality, this paper extends the mixing technique in TSM to the Transformer based backbone architecture, to obtain a space-time mixing attention in Video Transformer. This combination is straightforward and shown to be effective for video recognition.

The paper is well written and well organized.

Using the proposed methods, the paper achieves similar performance with much lower computation costs on Kinetics-400, Something-Something v2, and Epic Kitchens datasets.

Concerns:
1. The experimental results are not sufficient enough. This paper only presents one version of the X-ViT model with different input frames but did not verify the effectiveness of the proposed approach on different sizes of models.
2. The novelty is somewhat limited. This paper is more like a combination of TSM and Vision Transformer.
3. In Table 1, a larger temporal window size would lead to worse performance on video recognition. This may limit the application scenarios and hurt the generality of the proposed approach.
4. Number of parameters and speed of different models should be compared.
5. Other approaches could benefit from using more temporal clips, such as 4x3. But the presented approach could only achieve slightly better when using more temporal clips. Please explain why.

**Time Spent Reviewing:**

5

---

> ### Author Response · Authors · 2021-08-10
> **Response to reviewer 1 (C6rH)**
>
> **R1.1:** On insufficient experimental results because different model sizes were not tested.
>
> **A1.1:** We respectfully disagree for the following 3 reasons:
>
> 1. Firstly, we do provide an experiment illustrating the effectiveness of our approach using different model sizes, please see Table 4. There we show that a larger model adds $+1.2\%$. In general, most papers show that with bigger models better results are obtained. Because this is more or less an expected result, we preferred to use our resources to run other experiments as explained below (note that typically bigger models require hyper-parameter optimization and also take longer to train).
> 2. We ablate **all**  components of our model (e.g. different window sizes, SA position, space-time mixing, resolution, temporal attention, summary token, number of input frames) in Tables 1, 2 and 3.
> 3. We compare against the state-on-on-art space-time attentions [1,3] and prior work in video recognition on **the 4 most popular** video recognition datasets: Kinetics-400, Kinetics-600 (in the supplementary material), SS-v2 and Epic Kitchens.
>
> Overall, we have conducted a very large number of experiments. We believe these results clearly show the effectiveness of our method and how each individual component impacts the accuracy.
>
> **R1.2:** On novelty being somewhat limited.
>
> **A1.2:** Our paper is not Transformer plus TSM or (more precisely) Transformer plus "shift trick" (as the zero flop shift trick was originally proposed in [42]). We would like to make 3 points here:
> 1. Firstly, by applying the "shift trick", not as described in our method, but let's say to the feature tensor prior to the application of the self-attention layer (2nd row of Table 2(c)) does not result in a mathematically sound approximation to space-time attention. In contrast, we propose a valid approximation to space-time attention as outlined in Eq. 10 and Fig. 1 (d). This is derived by firstly making the temporal local approximation of Eq. 7 and then appropriately applying the "shift trick" to the key and values in order to obtain an efficient implementation of Eq. 8 (the whole process is described in L120-145).
> 2. Secondly, we would like to draw your attention again to Table 2(c). The table clearly shows that is absolutely essential to use the proposed method, based on a mathematically sound approximation to space-time attention, in order to achieve the highest possible accuracy. For example, simply mixing the feature tensor as described above is more than 1\% worse (2nd row of Table 2(c)). Other ways of applying the space-time mixing give even worse results.
> 3. Finally, another point that we want to make is that we show that our model in terms of joint *space-time local* attention is significantly more accurate than factorized versions of self-attention ([1, 3]) which might be ineffective when there is camera or object motion and there is spatial misalignment  between frames.
>
> **R1.3:** On "a larger temporal window size would lead to worse performance on video recognition. This may limit the application scenarios and hurt the generality of the proposed approach.".
>
> **A1.3:** Thank you for bringing up this point. We would like to make two points: Firstly, the local temporal window approach is not so limiting: as explained in L124-127 after a few transformer layers the whole clip is effectively covered. Secondly, and as also R2 very correctly pointed out, this value was found to be  optimal for SSv2 and, in general, for the short-video action recognition benchmarks we have considered. However, on other datasets different values might be needed. We will make sure to clarify this in the revised version of our paper. Thirdly, we believe that our results on all standard action recognition benchmarks (Kinetics-400, Kinetics-600 (in the supplementary material), SS-v2 and Epic Kitchens) convincingly show that our method significantly outperforms other existing approximations to space-time attentions [1,3] at least on these particular datasets.
>
> **R1.4:** On reporting number of parameters and speed.
>
> **A1.4:** We already report the computational complexity as measured in FLOPs for all the methods where we could find this data. We believe this is a good proxy metric and is one that is widely adopted by the community as the actual speed is subject to implementation. We also report in practical terms, as measured using PyTorch, the speed of our model with and without the addition of our local attention (L279-L284) showing that our model has nearly the same latency as a spatial-only attention video transformer! Following your suggestion we will also report the number of parameters. Bellow we list the number of parameters for a few methods from Table 6. Our approach uses fewer or equal number of parameters when compared with the best models from [1,3] that also use transformer architectures.
>
> | Method      | Parameters |
> | ----------- | ----------- |
> | STM      | 24M      |
> | TEA   | 25.6M       |
> |  i3D+NL   | 51.8M       |
> |   TSM R50   | 25.6M       |
> |   SlowFast-R101-8x8   | 62.8M       |
> |   X3D-XXL   | 20.3M       |
> |   ViVit-L  | ~312M       |
> |   TimeSformer  | 121M       |
> |   XViT (Ours)  | 92M       |
>
> Finally, we will also include the run-time for the methods that have publicly available implementations. We are using the metric proposed by R3(fqvk) that is to measure the number of videos processed per second. The number of frames and clips required to produce the prediction for a video follows the setting described by each method. The measurements follow the same setting as in the manuscript (8 V100 GPUs, batch size of 128 clips, averaged over 100 iterations). Please see bellow a few results. Notice that our method is significantly faster.
>
> bLVNet: 7.5videos/sec; TEA: 2.4videos/sec; i3D-NL ResNet101: 0.37videos/sec; SlowFast (8f): 0.69videos/sec; TimeSformer: 0.69videos/sec; ViVit: 0.66videos/sec; XViT (Ours): 12 videos/sec;
>
> **R1.5:**  On accuracy using more clips.
>
> **A1.5:**  We are sampling the frames as in TimeSformer [3], uniformly across the entire video. So similarly to [3], one temporal view covers the entire video. Using 2 temporal views did improve the accuracy in some of our experiments but not dramatically, see Fig. 1 of supplementary. Beyond 2 we did not observe any further improvements. However, this is actually an advantage of our approach that can work using less temporal crops to match and outperform significantly heavier approaches! In the 4x3 setting each clip covers only a portion of the video, therefore more clips are needed to capture the entire video.

---

> ### Author Response · Authors · 2021-08-28
> **Additional experiments and empirical evidence**
>
> In addition to our previous response we did manage to conduct the model size related experiments you requested. Futhermore, we provide clearer empirical evidence backing the novelty of our approach.
>
> ### 1. Experiments with different model sizes
>
> We agree that providing results with different model sizes is useful so we trained more models and included their accuracy in the updated version. Additional results that show the accuracy across different model sizes (Tiny, Small, Base, and Large) can be seen below (please note that some of them were already present in the initial version of the manuscript in Table 4.).
>
> Note that we report results for 4 different model sizes (Tiny, Small, Base, and Large) which is already more sizes than Timesformer [1] (they report S, B, L) and ViVit [3] (they report B, L, H). Moreover, compared to Timesformer [1] our S-model is just 2% worse than our B-model whereas in [1] the drop is 5%. Finally, we also report results with different patch sizes.
>
> | Method    | Top-1 | Top-5 |
> |-----------|-------|-------|
> | XViT-T/16 | 54.7 | 82.8 |
> | XViT-S/32 | 57.0 | 84.6 |
> | XViT-S/16 | 61.1 | 88.0 |
> | XViT-B/32  |59.8 | 87.4  |
> | XViT-B/16   | 63.4 | 89.6  |
> | XViT-L/32   | 61.0 | 88.3 |
> | XViT-L/16 | 64.1 | 89.9 |
> *Table 1. Accuracy of XViT for different model sizes and patch sizes.*
>
> ### 2. Empirical evidence of the differences with prior work
>
> In addition to the previous explanations, here we empirically show the superiority of our approach against a TSM-like adaptation for ViT.
>
> A straightforward application of the the shift trick to ViT, which from now on we will call TSM-ViT, can be described by the following Equations (note that this has not been described in some other paper before):
>
> $\mathbf{Z}^{l} = \texttt{SHIFT}(\mathbf{Z}^{l}) $
>
> $\mathbf{Y}^{l}  =  \textrm{MSA}(\textrm{LN}(\mathbf{Z}^{l-1})) + \mathbf{Z}^{l-1},$
>
> $\mathbf{Z}^{l+1}  = \textrm{MLP}(\textrm{LN}(\mathbf{Y}^{l})) + \mathbf{Y}^{l}. $
>
> Note that the differences between TSM-ViT and our model (XViT) are:
> 1. TSM-ViT does not perform a mathematically sound approximation to the full space-time attention as XViT does (described in our paper in **L118-L146** and **Eqs. (7)--(10)**).
> 2. TSM performs simple temporal average pooling for temporal aggregation. Instead we propose two new forms of aggregation: "Temporal Attention aggregation" (see **L147-156**) and "Summary Token" (see **L157-166**).
>
> The above model **was already evaluated in our paper in Table 2(c) second row**. For clarity, we report its accuracy in the following table together with XViT. Note that the first row in the table: *TSM-ViT (with final aggregation as proposed in TSM)* was not provided in the original submission.
>
> | Method    | Top-1 |
> |-----------|-------|
> | TSM-ViT (with final aggregation as proposed in TSM) | 59.8\% |
> | X-ViT (with final aggregation as proposed in TSM) | 61.6\% |
> | TSM-ViT (with final aggregation as proposed in L147-156) | 62.1\% |
> | X-ViT  (with final aggregation as proposed in L147-156) | 63.4\% |
> *Table 2. Comparison between XViT and "TSM-ViT". ViT-B/16 was used for all models.*
>
> We believe that these results clearly show that these two approaches are different. This is to be expected as XViT is the only one that performs a mathematically sound approximation to the full space-time attention. Moreover, we see that the proposed temporal attention aggregation is much more effective than the simple temporal pooling proposed by the original TSM paper.
>
> Should additional clarification be required, we would be more than happy to provide it. We are looking forward to your reply.

---

### Author Response · Authors · 2021-08-10
**Overall response**

We would like to thank all Reviewers for their time and effort to provide feedback for our work. We hope that we address all concerns raised below. We would be also more than happy to come back with more clarifications should they be further requested by the Reviewers.

---

### Decision · Program_Chairs · 2021-09-27

**Decision:**

Accept (Poster)

**Comment:**

The authors propose a novel transformer architecture for video recognition whose performance scales linearly with the number of frames (as opposed to the usual quadratic scaling). To achieve this the authors restrict the time attention to a local temporal window, and introduce an efficient space-time mixing procedure. The proposed approach offers competitive results in terms of accuracy/flops tradeoffs on several popular video recognition benchmarks.

The paper was reviewed by 4 expert reviewers and received borderline ratings. The rebuttal managed to address points raised by most reviewers, who found it a valuable contribution to the community. One reviewer maintains the view that the work lacks novelty in terms of technical contributions, and that more ablation studies are necessary. After considering the manuscript, the reviews, and the discussion, I felt that the work should be accepted for publication and that a minor revision is sufficient to address the raised criticisms.